# Deprivation and Aspiration Strains as Function of Mental Health Among Chinese Adults: Study of a National Sample

**DOI:** 10.3390/ijerph21121598

**Published:** 2024-11-30

**Authors:** Jie Zhang, Lulu Zhao, Dorian A. Lamis

**Affiliations:** 1School of Public Health, Shandong University, Jinan 250012, China; 2Department of Sociology, State University of New York Buffalo State University, Buffalo, NY 14222, USA; 3Department of Sociology, Central University of Finance and Economics, Beijing 100081, China; zhaolulu2019@163.com; 4Department of Psychiatry and Behavioral Sciences, Emory University School of Medicine, Atlanta, GA 30322, USA; dorian.lamis@emory.edu

**Keywords:** deprivation strain, aspiration strain, mental health, China

## Abstract

*Background:* China is in a period of rapid transformation of economic and social development. The imbalance in the distribution of social benefits, focusing on adjustment and reorganization, has led to an increase in relative deprivation. Studies have shown that relative deprivation leads to a decline in personal mental health. *Methods:* We used the national data in the Chinese General Social Survey (CGSS) collected in 2015 (N = 10,702; male 46.89%) to estimate the relationship between deprivation and aspiration strains and mental health. Covariates included age, gender, education level, and marital status. *Results:* The results show the mechanism of how relative social and economic status affects the mental health of residents through deprivation strains and aspiration strains. It was found that deprivation strains and aspiration strains are both negatively correlated with mental health, even when controlling for the confounding variables. *Conclusion:* Mental health can be improved, and suicide rates may be decreased by vigorously developing the economy, promoting social equity and justice, and strengthening psychological counseling for the general population.

## 1. Introduction

With the stable and orderly development of the social economy in the past several decades, people’s living standards and social medical conditions have been greatly improved. Thus, the health level and life expectancy of residents have also greatly improved. According to the 2018 World Health Organization (WHO) report, the average life expectancy of Chinese people has reached 76.4 years, an increase of 2.9 years compared to 2016. Based on this, many scholars have speculated that with the further development of the economy, the gap between people would be effectively narrowed, and health inequality would be alleviated; however, this inference has not been supported in the scientific literature. On the contrary, researchers have found that with the sustainable development of society and the economy, health inequality has not been alleviated but has begun to show an expanding trend [1] In both developed or developing countries, vast differences exist in the health status of people under different conditions. Empirical evidence demonstrates a persistent positive correlation between socioeconomic status and health outcomes, with individuals of higher socioeconomic status experiencing significantly better health indicators compared to their lower-status counterparts [2]. This systematic health disparity poses a substantial challenge to equitable social development [3].

Throughout its extensive history, China has been shaped by Confucian cultural traditions, which have fundamentally influenced societal values and perceptions of inequality [4]. A distinctive cultural paradigm has emerged wherein Chinese society exhibits greater concern about resource distribution inequity than absolute scarcity, and social instability rather than population density [5]. Individuals may experience psychological deprivation despite possessing abundant social resources when they perceive disparities between themselves and their reference groups, a phenomenon that manifests independently of actual economic losses. Deprivation strains are an internal psychological experience, primarily from people thinking that they are in a lower position in society compared to the horizontal or vertical reference group. The perception that others have secured economic and social advantages that one rightfully deserves can trigger feelings of social subordination, leading to psychological distress and diminished mental health outcomes [6].

As an important part of social stratification, health inequality has received increased attention in social science. Recently, a burgeoning body of literature has emerged investigating the relation between health and the absolute objective economic status [7,8]. Researchers believe that there is a strong association between health and an individual’s absolute objective economic status, and that a low socioeconomic status is significantly associated with poor mental health outcomes [9]. Few scholars, however, are concerned about the impact of relative income or personal deprivation strains on resident health, especially if there was an explanatory effect in the process of socioeconomic status on health. In the current study, we will examine subjective socioeconomic status, deprivation strains, and aspiration strains.

The strain theory of mental disorders is applied to the current research, which proposes that psychological strains usually precede mental disorders [10]. According to this theory, strain is also an uncoordinated pressure, which refers to the pressure generated by two or more opposing concepts or social experiences internalized in the individual’s psychology. Strain is different from ordinary pressure; the formation of strain must have two or more pressure sources [11]. There are four different sources of strain: strain between different values, conflict between expectations and reality, strain between expectations and other deprivation strains, and strains due to the inability to deal with crisis. Strains will make it difficult for individuals to adapt to society and could potentially cause harm to others. There are two kinds of strains in this article: (1) aspiration strains caused by the difference between desire and reality; (2) deprivation strains including poverty or being relatively poor [12]. Stress caused by contradictions and competition in personal life leads to nervous emotions and intense emotion that can potentially lead to suicidal behavior. Relative deprivation and aspiration deprivation can also produce stress and tension, threatening people’s health. In addition, this theory has been tested in the United States and supports the relationship between the four sources of stress, well-being, and mental health. It has been used in China to explore the effects of four kinds of strains on a variety of groups [13].

By systematically reviewing the literature between subjective socioeconomic status, deprivation strains, aspiration strains, and mental health, based on data from the 2015 China Comprehensive General Social Survey (CGSS), in this study we attempt to answer the following two questions: First, will the relative socioeconomic status and deprivation strains of individuals and the resulting distortions have an impact on health? If so, to what extent can it explain the inequality of health outcomes? Second, how does socioeconomic status affect residents’ mental health through deprivation strains and aspiration strains?

## 2. Materials and Methods

### 2.1. Data and Samples

The data in this article are based on a sample of the Chinese General Social Survey (CGSS) led by the National Survey and Research Center of Renmin University of China in 2015. The CGSS was formally launched in 2003 and employs rigorous probability sampling methods to collect data on social trends, living conditions, and public attitudes across urban and rural China. The survey data were published on the official website of China National Survey Database (CNSDA) on 1 January 2024. Multi-layer stratified sampling was used in each stage of the survey and sample, including the main sampling unit in the area, the secondary sampling unit in the residential area, and the third sampling unit in the residents’ committee or village committee. From each sampled household, a qualified person aged 18 years or older was randomly selected as the survey subject [14,15]. After rigorous screening, we obtained 10,702 effective respondents from the 2015 survey, including 5018 male respondents and 5684 female respondents. The survey covers diverse topics including demographics, education, employment, health, social networks, and values, making it a crucial resource for researchers studying social phenomena in China. For more detailed information about the CGSS, see http://cgss.ruc.edu.cn (accessed on 10 March 2020).

### 2.2. Measurement

Our dependent variable is the mental health of the residents, using subjective self-assessment health indicators commonly used in sociology. According to the level of mental health, we divided mental health into five categories: very healthy, relatively healthy, healthy, relatively unhealthy, and very unhealthy. The most relevant item in the CGSS2015 questionnaire for our analysis asks respondents how often they feel depressed, or depressed in the past four weeks. The higher the frequency of depression, the lower the mental health. The answer is coded according to mental health from high to low, ranging from 1 (very healthy) to 5 (very unhealthy). Mental health is a ranking order variable, and we can treat it as a continuous variable (1–5), using *t*-test for the gender comparison. The higher the score, the worse the mental health.

In this study, we examine relative deprivation and deprivation strains as core independent variables. Among them, relative deprivation and deprivation strains are divided into three types: high, middle, and low. The operational problem of relative deprivation in CGSS2015 is: compared with people of the same age, what is your socioeconomic status? The response options are coded as 1 = high, 2 = middle, and 3 = low. In addition, the operational problem of torque deprivation in CGSS2015 is: Compared to three years ago, what is your socioeconomic status? The respondent’s answer options are coded 1 = high, 2 = middle, and 3 = low. Relative deprivation and deprivation strains are ranking variables. We used the *t*-test to compare genders and treat them as continuous variables (1–3). Given that the associations among deprivation strains, aspiration strains, and mental health is our primary theoretical concern, we controlled for other relevant variables in all statistical analysis. For the interviewees, we created a dummy variable for gender (male = 1), a dummy variable for marital status (currently married = 1), and included age, physical health status, and education level in the model [7].

### 2.3. Analysis Strategy and Model

STATA 15 was used for data analysis. We employed *t*-tests or chi-square tests to compare the differences between categorical and continuous variables between the two groups. The level of statistical significance was set at *p* < 0.05, indicating that results were considered statistically significant when the probability of obtaining such results by chance was less than 5%. Regression analysis was carried out on the variables of deprivation strains, aspiration strains, and mental health. At present, the relationship between these three variables can be studied with different methods and models, such as structural equation models and regression models. Considering the characteristics of explanatory variables, the multivariate linear regression model is more suitable for estimating the mental health of Chinese residents in 2015:y = a + β_1_x_1_ + β_2_x_2_ + … + β_i_x_i_ + ε (i = 1, 2, …, n)(1)

## 3. Results

### 3.1. Descriptive Statistics

The number of males in the sample is 46.89% and the number of females is 53.11% of the total sample, which is consistent with previous research. Table 1 shows the average (standard deviation) or frequency (percentage) of each variable and its gender difference comparison. It can be concluded that the average score of mental health for the entire sample is 3.84, and there is a significant difference between male and female respondents (*p* < 0.001). The average age in the sample is 50.40 (SD = 16.87). Compared with men, women tend to be younger (M = 50.36, SD = 16.80). In addition, in our sample, compared to men (M = 3.69), the health status of women is worse (M = 3.54), and the education level of women (M = 3.20) is also lower than that of men (M = 2.82). These differences are statistically significant (*p* < 0.001). Moreover, there are 22.17% currently unmarried people and 77.83% currently married people (N = 10,702), which may be related to the fact that our survey subjects are adults. In terms of age, deprivation strains, aspiration strains, and marital status, no statistically significant gender differences were found.

### 3.2. Binary Correlation Analysis of Various Factors and Mental Health

The results of the binary correlation analysis (see Table 2) show that deprivation strains, aspiration strains, age, physical health, education level, and gender are all significantly related to mental health (*p* < 0.001). Both deprivation and aspiration strains are negatively correlated with mental health, and higher levels of deprivation and aspiration strains both lead to lower mental health status. The older you are, the better your mental health is in the current sample. The better one’s physical health and education, the better the mental health. In addition, men have better mental health than women in our sample.

### 3.3. Multiple Regression Analysis of Various Factors and Mental Health

We constructed multiple regression to analyze factors related to mental health (see Table 3). The multiple linear regression model was adopted for analysis, and the following is the model:y = a + β_1_x_1_ + β_2_x_2_ + … + β_i_x_i_ + ε (i = 1, 2, …, n)(2)

There are three models in this study: Model 1 is the relative deprivation model, Model 2 is the desire deprivation model, and Model 3 is the overall deprivation model.

The independent variables in Model 1 are deprivation strains; the dependent variables are mental health status; age, gender, marital status, education level, and physical health status were added to the model as control variables. The results reveal that the regression equation is significant and explains 22.4% of the total variance. Through analysis, we know that in Model 1, the impact of gender on mental health has no statistical significance (β_2_ = −0.017, *p* > 0.001). The impact of age (β_1_ = 0.005, *p* < 0.001), marital status (β_3_ = 0.049, *p* < 0.001), physical health (β_4_ = 0.374, *p* < 0.001), education level (β_5_ = 0.083, *p* < 0.001), and relative deprivation (β_6_ = −0.134, *p* < 0.001) on mental health are statistically significant.

In Model 2, the effects of age, marital status, physical health, education level, and aspiration strains on mental health are statistically significant. Among the individual’s physical variables (age, gender, marital status, physical health status), the older the age, the better the mental health status (β_1_ = 0.006, *p* < 0.001), and currently married individuals have better mental health than unmarried people (β_3_ = 0.054, *p* < 0.001). The better the physical health, the better the mental health (β_4_ = 0.383, *p* < 0.001). In addition, we also found that women’s mental health status is worse than men’s mental health, although the *p* value of the gender variable is not statistically significant (β_1_ = −0.013, *p* > 0.001). Among the individual’s sociocultural variables (level of education), those with a high level of education have higher mental health status, and this effect is statistically significant (β_5_ = 0.096, *p* < 0.001). In the variables of economic status, there is a negative correlation between the individual’s aspiration deprivation and mental health. The stronger the individual’s desire deprivation, the worse the individual’s mental health will be (β_7_ = −0.076, *p* < 0.001). Analysis shows that the model can explain 22.1% of the total variance.

Model 3 shows that age, marital status, physical health, education, deprivation, and aspiration strains have significant effects on mental health. The specific performance is: the older the age, the better the mental health status (β_1_ = 0.005, *p* < 0.001); compared with currently unmarried people, a currently married person’s mental health status is better (β_3_ = 0.048, *p* < 0.001); physical health status positively correlates with mental health status. The better the physical health status, the better the mental health status (β_4_ = 0.372, *p* < 0.001); the higher the education level, the better the mental health status (β_5_ = 0.084, *p* < 0.001). There is a negative correlation between deprivation and aspiration strains and mental health (β_6_ = −0.121, *p* < 0.001 β_7_ = −0.057, *p* < 0.001). The R^2^ of model 3 is 22.5%, which is higher than the R^2^ of the first two models. This shows that the overall relative deprivation model fits better than the relative deprivation model of a certain dimension alone.

## 4. Discussion

Although the relationship between deprivation stress and mental health has been extensively studied [14], the mechanisms that lead to this association have not been fully understood. In this study, we focused on the association between adult deprivation strain, aspiration strain, and mental health in Chinese individuals. Compared to western civilizations, in this study, we utilize a sample of 10,702 Chinese adults, which improves the potential for generalizing the results to other populations. In addition, compared to previous studies on the relationship between objective socioeconomic status indicators and mental health, we investigated how relative deprivation strain and aspirational strain—psychological constructs capturing temporal and social comparative processes—impact mental health.

The results reveal that the older the age, the better one’s mental health status, which is consistent with previous research [16]. One explanation for this finding may be that as people get older, an individual’s knowledge of themselves is more accurate, and the individual is able to choose the appropriate reference group, avoiding the excessive anxiety which results from the incorrect reference group selection. In addition, marital status is an important family factor, which is positively related to mental health. It is universally acknowledged that in a healthy marital relationship, couples will help each other when they encounter difficulties not only financially, but also spiritually and through daily care. These will enhance the individual’s ability to cope with life’s difficulties and resist strains. Whether in single factor analysis or multiple linear regression analysis, the positive effect of physical health on mental health was statistically significant. Some researchers have found that the health self-assessment status is heavily influenced by subjective socioeconomic status and other factors, rather than the objective income of residents [17]. Moreover, as supported by several studies [18], the higher the level of education, the better the mental health status. This may be due to the enhanced socioeconomic status and the higher level of education in China. Indeed, previous research has shown that subjective social status in a low-income minority sample is related to mental health [19,20]. In addition, the higher the education level, the more cultural capital and the stronger the ability to cope with a life crisis, which helps to maintain an improved mental health status. While controlling for other variables, our results revealed that the higher the deprivation and aspiration strains, the worse the mental health. The empirical evidence derived from our CGSS analysis corroborates existing findings from both cross-regional studies and local Chinese research, demonstrating consistency across multiple empirical contexts and suggesting generalizability of these conclusions. Finally, it is worth noting that, unlike previous studies, given that we controlled for other variables, the influence of gender factors on individual mental health is not statistically significant.

Our findings also answer the two questions raised at the beginning of the paper: First, compared with the impact of objective socioeconomic conditions on health, individuals will have a certain deprivation by comparing their socioeconomic status with surrounding reference groups and their past selves, causing deprivation strains and aspiration strains. This process affects the individual’s mental health. Second, through univariate analysis, we found that both deprivation strains and aspiration strains will have a negative impact on an individual’s mental health. In the current multiple regression analysis, we found that while controlling for covariates (e.g., age, gender, marital status), deprivation strains and aspiration strains were both significantly negatively correlated with mental health. Model 3 explained 22.5% of the variance affecting mental health, which is a high degree of interpretation in the social sciences. In addition, the results of the study show that deprivation strains have a greater impact on mental health than aspiration strains. The findings suggest that psychological strain arising from lateral social comparisons with peer reference groups exhibits stronger associations with mental health outcomes than strain generated through temporal self-comparison. Third, some scholars have shown that there is a correlation between depression and subjective relative deprivation through scales. Subjective relative deprivation is closely related to depression, in which depression cognition, emotional regulation, and subjective stress can potentially lead to depression. The regulation of negative automatic thoughts attenuates the relationship between subjective relative deprivation and depressive symptoms [21]. Based on the existing data and in line with the strain theory, this study advances an alternative explanatory mechanism. The comparison of individual subjective socioeconomic status supports the impact of two of the four sources of strains on mental health: the source of strain between expectation and others (deprivation strains) and the source of strain between expectation and reality (aspiration strains) emergence. Thus, the findings demonstrate that both sources of strain exert significant adverse effects on mental health outcomes, with their concurrent impact being associated with particularly diminished psychological well-being.

## 5. Limitations

The main limitation of this study is the use of a cross-sectional design. Although the design can be used to identify statistical associations, it does not provide information about whether the presumed cause precedes the effect, or whether the association between deprivation strains, aspiration strains, and life satisfaction is driven by a third factor. To address this limitation, we can conduct longitudinal research in the future, so that the relevant factors may be explained more effectively, and their impact can be predicted more accurately. Moreover, future researchers should investigate the association among the primary study variables in other cultures. In addition, we investigated the sense of deprivation of the Chinese people as a whole, and focused on issues at the macro level; therefore, it is impossible to determine the micro-mechanism of deprivation pressure. Future studies should address this limitation with research at the micro-level.

## 6. Conclusions

The results of this study clearly show the negative effects of deprivation and aspiration strains on individual’s mental health. This effect is significant even in the presence of control variables. Deprivation and aspiration strains are a common social phenomenon in any society, so it is impossible to eliminate them. However, it is necessary to ease the deprivation strains and aspiration strains of the people. In addition, China is in a period of social transformation [22]. Economic disparities and social stratification have exhibited accelerating expansion, demonstrating an increase in people’s sense of deprivation strains, such as the generation of hatred for wealth, revenge, populism, etc. Thus, it is urgent to alleviate the public’s sense of deprivation, as the income gap is a source of deprivation strains. While economic development and wealth redistribution mechanisms may mitigate these disparities, the preservation of social equity remains paramount for addressing these systemic challenges [23,24]. In order to foster equitable distribution of economic advancement and enhance collective psychological well-being, it is essential to facilitate appropriate social comparison frameworks that may attenuate societal discontent and alleviate social anger.

## Figures and Tables

**Table 1 ijerph-21-01598-t001:** Characteristics of the sample and the gender differences for major variables.

Variable	Total (N = 10,702) M ± SD/f (%)	Male (N = 5018) M ± SD/f (%)	Female N = 5684 M ± SD/f (%)	*t*	*p*
Age	50.40 ± 16.87	50.45 ± 16.95	50.36 ± 16.80	0.29	0.775
Deprivation Strains	2.29	2.27	2.30	−2.52	0.012
High	519 (4.79%)				
Middle	6670 (61.61%)				
Low	3637 (33.60%)				
Aspiration Strains	1.76	1.75	1.76	−0.73	0.468
High	3754 (34.55%)				
Middle	5995 (55.18%)				
Low	1115 (10.26%)				
Mental Health	3.84	3.90	3.79	6.05	<0.001
Very unhealthy	126 (1.16%)				
Unhealthier	749 (6.91%)				
Healthy	2602 (24.00%)				
Healthier	4613 (42.54%)				
Very healthy	2753 (25.39%)				
Marital status	1.79	1.78	1.78	1.00	0.316
Currently unmarried	2409 (22.17%)				
Currently married	8455 (77.83%)				
Physical Health	3.61	3.69	3.54	7.21	<0.001
Very unhealthy	341 (3.14%)				
Unhealthier	1603 (14.76%)				
Healthy	2349 (21.63%)				
Healthier	4229 (38.94%)				
Very healthy	2337 (21.52%)				
Education	3.00	3.20	2.82	15.54	<0.001
Illiteracy	1472 (13.55%)				
Primary School	2616 (24.08%)				
Junior Middle school	3073 (28.29%)				
High School	1957 (18.01%)				
Undergraduate	1643 (15.12%)				
Postgraduate and Above	103 (0.95%)				

**Table 2 ijerph-21-01598-t002:** Bivariate analyses of major variables.

	Mental Health
Deprivation Strains	*r* = −0.31, *p* < 0.001
Aspiration Strains	*r* = −0.14, *p* < 0.001
Age	*r* = −0.07, *p* < 0.001
Physical Health	*r* = 0.39, *p* < 0.001
Education	*r* = 0.15, *p* < 0.001
Gender (Female)	*r* = −0.11, *p* < 0.001
Marital status	*r* = 0.03, *p* > 0.001

**Table 3 ijerph-21-01598-t003:** Multiple regressions of mental health.

	Model 1 N = 10,801	Model 2 N = 10,838	Model 3 N = 10,801
Age	0.005 ***	0.006 ***	0.005 ***
Gender (Female)	−0.017	−0.013	−0.017
Marital status	0.049 ***	0.054 ***	0.048 **
Physical Health	0.374 ***	0.383 ***	0.372 ***
Education	0.083 ***	0.096 ***	0.084 ***
Deprivation Strains	−0.134 ***		−0.121 ***
Aspiration Strains		−0.076 ***	−0.057 ***
Constant	2.233 ***	1.932 ***	2.302 ***
R-squared	0.224	0.221	0.225

*** *p* value is significant at the 0.01 level (two-tailed), ** *p* value is significant at the 0.05 level (two-tailed).

## Data Availability

The data are publicly available: http://cgss.ruc.edu.cn (27 November 2024).

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
