# Peer review of "Deprivation and Aspiration Strains as Function of Mental Health Among Chinese Adults: Study of a National Sample"

_ijerph, 2024, doi:10.3390/ijerph21121598_

Round 1
Reviewer 1 Report
Comments and Suggestions for Authors
Dear Authors,
the manuscript submitted for review addresses an extremely important issue for Public Health, i.e. the issue of mental health. Nevertheless, I believe that the content of the manuscript is not entirely consistent with the topic. Mental health in poor condition or, for example, depression does not always mean suicidal tendencies. I propose to verify the topic of the article by limiting it to mental health, because generally there is little content in the manuscript on the subject of suicidal behavior, and above all, the study itself does not include people who have had such experiences (lack of consistency between the topic and the research project). The respondents probably did not even declare such tendencies, only bad or very bad mental health. In addition, I propose to edit the introduction and move part of the content to the discussion, which lacks scientific dialogue and references to other studies in the studied area (e.g. lines 64-126). I also suggest removing information from the introduction that constitutes a description of the "material". As for the research method, please supplement the content "material and method" with the level of significance adopted in the study (p Value). Unfortunately, in my opinion, the conclusions also require improvement. They are partly inconsistent with the presented results, e.g. they deal with the low socio-economic status of rural residents (?). The conclusions are also not the place for a scientific discussion. I also have doubts about the literature. Some of the items require updating.
Best regards
Author Response
Reviewer comments 1
1) The manuscript submitted for review addresses an extremely important issue for Public Health, i.e. the issue of mental health. Nevertheless, I believe that the content of the manuscript is not entirely consistent with the topic. Mental health in poor condition or, for example, depression does not always mean suicidal tendencies. I propose to verify the topic of the article by limiting it to mental health, because generally there is little content in the manuscript on the subject of suicidal behavior, and above all, the study itself does not include people who have had such experiences (lack of consistency between the topic and the research project). The respondents probably did not even declare such tendencies, only bad or very bad mental health.
Response: We thank this reviewer for this bringing this to our attention and for the suggestion to focus on mental health as opposed to suicide. We have changed the topic of the article by removing “suicidality.”
2) In addition, I propose to edit the introduction and move part of the content to the discussion, which lacks scientific dialogue and references to other studies in the studied area (e.g. lines 64-126).
Response: We deleted two large paragraphs from the introduction as suggested by the reviewer. We also moved parts of the introduction to the discussion as recommended.
3) I also suggest removing information from the introduction that constitutes a description of the "material".
Response: We have removed this from the text.
4) As for the research method, please supplement the content "material and method" with the level of significance adopted in the study (p Value).
Response: We have made this change
5) Unfortunately, in my opinion, the conclusions also require improvement. They are partly inconsistent with the presented results, e.g. they deal with the low socio-economic status of rural residents (?). The conclusions are also not the place for a scientific discussion. I also have doubts about the literature. Some of the items require updating.
Response: We have revised the conclusion section.
Reviewer 2 Report
Comments and Suggestions for Authors
This manuscript presents results of a sample of Chinese participants of the national Chinese General Social Survey. This study is intended to assess the influences of deprivation and aspiration strains on mental health, including suicidal thoughts. The hypotheses and background of the study are derived from the Strain Theory of suicide and mental health disorders, with social and psychological components to explain the occurrence of suicide or mental disorders in a culture or individuals. The study supports the influence of these strains in predicting mental health, with deprivation strains more influential than aspiration strains, along with other forces (e.g., physical health, income). The authors provide some limitations of the investigation as well as future research needs and cultural changes that are implied by the study findings.
Among the strengths of the study are the sizeable number of participants included in the sample (over 10,000). Relatedly, the selection of the sample from the national-level Chinese General Social Survey provides a strength with respect to the representativeness of the participants. An additional strength is the foundation of a theoretical base to the study in Strain Theory.
There are possible issues that might be addressed.
(1) It would be informative to provide a bit more descriptive and substantive information about the Chinese General Social Survey. For example, what proportion of the original, complete sample does the large sample selected here represent? The slightly larger inclusion of more women than men is indicated as reflecting the larger complete survey, but women were younger than men in the sample studied here (true in the complete survey also?). The few findings associated with gender differences could conceivably be influenced by the gender makeup of the age groupings. How often is the Chinese GSS conducted? Has there been one since 2015?
(2) In the Limitations paragraph and its forward look to future research, there could be a suggestion regarding the investigation of these issues in other cultures, with perhaps other patterns or findings related to cultural differences emerging.
(3) Miscellaneous/Minor issues – (a) In a few spots in the manuscript male pronouns are utilized (e.g., line 266). The substitution of a gender neutral term could be considered. (b) In at least a few cases (line 138 “data was”; line 138 “data is”) a singular verb is used for the plural subject “data.” (c) The Court (1981) reference listing in the References section (number 5) should be Court, SD (i.e., no “and”).
Comments on the Quality of English LanguageThe English language is quite good. There are some phrasing issues with articles for example that might be edited, though the article is clear as submitted.
Author Response
Reviewer comments 2
1) Among the strengths of the study are the sizeable number of participants included in the sample (over 10,000). Relatedly, the selection of the sample from the national-level Chinese General Social Survey provides a strength with respect to the representativeness of the participants. An additional strength is the foundation of a theoretical base to the study in Strain Theory.
Response: We thank this reviewer for commending the strengths of this study
2) It would be informative to provide a bit more descriptive and substantive information about the Chinese General Social Survey. For example, what proportion of the original, complete sample does the large sample selected here represent? The slightly larger inclusion of more women than men is indicated as reflecting the larger complete survey, but women were younger than men in the sample studied here (true in the complete survey also?). The few findings associated with gender differences could conceivably be influenced by the gender makeup of the age groupings. How often is the Chinese GSS conducted? Has there been one since 2015?
Response: We included additional information about the CGSS and added the official website that readers can access for more details.
3) In the Limitations paragraph and its forward look to future research, there could be a suggestion regarding the investigation of these issues in other cultures, with perhaps other patterns or findings related to cultural differences emerging.
Response: We have added this in the limitation section
4) Miscellaneous/Minor issues – (a) In a few spots in the manuscript male pronouns are utilized (e.g., line 266). The substitution of a gender neutral term could be considered. (b) In at least a few cases (line 138 “data was”; line 138 “data is”) a singular verb is used for the plural subject “data.” (c) The Court (1981) reference listing in the References section (number 5) should be Court, SD (i.e., no “and”).
Response: We have addressed these concerns in the manuscript.
5) Comments on the Quality of English Language. The English language is quite good. There are some phrasing issues with articles for example that might be edited, though the article is clear as submitted.
Response: We have improved the English language throughout the manuscript.
Round 2
Reviewer 1 Report
Comments and Suggestions for Authors
Dear Authors,
thank you for the corrections you made. However, I propose to consistently remove all content regarding suicide and suicidal thoughts from the text and focus only on mental disorders, i.e. remove:
- line 24 (keywords)
- lines 124-126
- lines 395-397.
Unfortunately, I still do not see in the description of the method information about the level of significance (p) adopted in the study, I only see "p" next to the obtained results.
To my knowledge, conclusions are not the appropriate place for citations, but this is just a suggestion.
Best regards
Author Response
Reviewer 1
Comment: I propose to consistently remove all content regarding suicide and suicidal thoughts from the text and focus only on mental disorders, i.e. remove:
- line 24 (keywords)
- lines 124-126
- lines 395-397.
Response: We have removed all content related to suicide as suggested by this reviewer’s specific instances.
Comment: Unfortunately, I still do not see in the description of the method information about the level of significance (p) adopted in the study, I only see "p" next to the obtained results.
Response: We have addressed this and included a statement on that the “the level of statistical significance was set at p < .05” in the “Analysis Strategy and Model” section
Comment: To my knowledge, conclusions are not the appropriate place for citations, but this is just a suggestion.
Response: Thank you for this suggestion. We have limited references in the Conclusion section unless necessary to cite a statement.
Reviewer 2 Report
Comments and Suggestions for Authors
This revised manuscript presents results of a sample of Chinese participants of the national Chinese General Social Survey. This study is intended to assess the influences of deprivation and aspiration strains on mental health.
A considerable revision of the text was made from the initial manuscript. The revision addresses issues raised by reviewers and the changes tighten the manuscript.
There are possible minor issues that might be addressed.
(1) An apparent inconsistency was observed regarded the findings for the relation of mental health and age. On line 232 it is stated that “The older you are, the worse your mental health.” However, line 257 says “the older the age, the better the mental health status.” And similarly on line 299 “The results reveal that the older the age the better one’s mental health status.” This seeming discrepancy might be addressed or explained.
(2) Miscellaneous Minor issues – (a) line 125, “psychological strains usually precedes mental disorder.” The plural “precede” would seem appropriate. (b) Line 219 omits a closing parenthesis after “16.80. (c) Line 224, the word “there” could be omitted given the editing changes made in this revision. (d) A singular verb is used for the plural subject “data” in line 162 (“data was”) (e) The first reference listed in the References section (number 1; Court (1981)) should be Court, SD (i.e., no “and”). This information is directly from the journal’s webpage, that is, “S D Court”.
Author Response
Reviewer 2
Comment: (1) An apparent inconsistency was observed regarded the findings for the relation of mental health and age. On line 232 it is stated that “The older you are, the worse your mental health.” However, line 257 says “the older the age, the better the mental health status.” And similarly on line 299 “The results reveal that the older the age the better one’s mental health status.” This seeming discrepancy might be addressed or explained.
Response: We have addressed this discrepancy to make everything consistent that the older one gets, the better their mental health
Comment: (2) Miscellaneous Minor issues – (a) line 125, “psychological strains usually precedes mental disorder.” The plural “precede” would seem appropriate. (b) Line 219 omits a closing parenthesis after “16.80. (c) Line 224, the word “there” could be omitted given the editing changes made in this revision. (d) A singular verb is used for the plural subject “data” in line 162 (“data was”) (e) The first reference listed in the References section (number 1; Court (1981)) should be Court, SD (i.e., no “and”). This information is directly from the journal’s webpage, that is, “S D Court”.
Response: We thank this reviewer for pointing out these minor issues. We have addressed them all.